# Epidural Steroid Injections for Low Back Pain: A Narrative Review

**DOI:** 10.3390/ijerph19010231

**Published:** 2021-12-26

**Authors:** Massimiliano Carassiti, Giuseppe Pascarella, Alessandro Strumia, Fabrizio Russo, Giuseppe Francesco Papalia, Rita Cataldo, Francesca Gargano, Fabio Costa, Michelangelo Pierri, Francesca De Tommasi, Carlo Massaroni, Emiliano Schena, Felice Eugenio Agrò

**Affiliations:** 1Unit of Anaesthesia, Intensive Care and Pain Management, Department of Medicine, Campus Bio-Medico University of Rome, 00128 Rome, Italy; g.pascarella@unicampus.it (G.P.); a.strumia@unicampus.it (A.S.); r.cataldo@unicampus.it (R.C.); f.gargano@unicampus.it (F.G.); f.costa@unicampus.it (F.C.); f.agro@unicampus.it (F.E.A.); 2Department of Orthopaedic and Trauma Surgery, Campus Bio-Medico University of Rome, 00128 Rome, Italy; fabrizio.russo@unicampus.it (F.R.); g.papalia@unicampus.it (G.F.P.); 3Integrated Sleep Surgery Team UCBM, Unit of Otolaryngology, Integrated Therapies in Otolaryngology, Campus Bio-Medico University of Rome, 00128 Rome, Italy; michelangelo.pierri@unicampus.it; 4Unit of Measurements and Biomedical Instrumentation, Department of Engineering, Campus Bio-Medico University of Rome, 00128 Rome, Italy; f.detommasi@unicampus.it (F.D.T.); c.massaroni@unicampus.it (C.M.); e.schena@unicampus.it (E.S.)

**Keywords:** epidural steroid injections, low back pain, lumbosacral radicular pain, disk herniation, canal stenosis, review

## Abstract

Low back pain represents a significant socioeconomic burden. Several nonsurgical medical treatments have been proposed for the treatment of this disabling condition. Epidural steroid injections (ESIs) are commonly used to treat lumbosacral radicular pain and to avoid surgery. Even though it is still not clear which type of conservative intervention is superior, several studies have proved that ESIs are able to increase patients’ quality of life, relieve lumbosacral radicular pain and finally, reduce or delay more invasive interventions, such as spinal surgery. The aim of this narrative review is to analyze the mechanism of action of ESIs in patients affected by low back pain and investigate their current application in treating this widespread pathology.

## 1. Introduction

Low back pain (LBP) and lumbosacral radicular pain are common causes of physical and mental morbidity and they are also a significant economic burden, causing an expenditure of more than USD 100 billion per year in the United States alone [1,2]. In the medical literature, low back pain is referred to as sciatica, lumbosacral radicular syndrome, lumbar radiculopathy, nerve root pain and nerve root entrapment/irritation, and is commonly described as a pain starting in the back and radiating to the legs. The etiological cause of low back pain is first represented by intervertebral disk disease. The pathophysiological changes involved in the intervertebral disk disease may lead to disk herniation or degenerative diseases, such as canal stenosis or chronic instability of the diseased segments. The most common cause of sciatica is the herniation of the nucleus pulposus, a component of the intervertebral disk in the lumbar region, which causes stenosis and inflammation [3,4]. Some estimate that sciatica caused by herniation of the lumbar disk has a prevalence of 9.8 out of 1000 [5], meaning that of all reported cases of sciatica it appears that 90% are caused by herniation of the lumbar disk [6].

Several nonsurgical medical treatments have been proposed for lumbosacral radicular pain, from lifestyle changes, exercise and physical therapy to analgesic local/oral drugs and epidural steroid injections (ESIs) [7,8]. The conservative management of LBP aims to delay or avoid surgery. As a matter of fact, LBP can improve spontaneously or with conservative treatment. Cases which do not respond to treatment are candidates for surgical intervention.

An ESI is a common and minimally invasive procedure, performed to successfully treat lumbosacral radicular pain, which has also proved its effectiveness in the treatment of back acute pain and leg symptoms. The injections are used to deliver steroids, and sometimes local anesthetics, to the epidural space, directly to the site that causes the pain using a caudal, interlaminar or transforaminal approach [9]. The epidural injection is a well-founded anesthetic and analgesic technique; moreover, nowadays, new technological devices can help anesthesiologists to learn and to administer it [10,11,12,13,14,15,16]. Even though it is still not clear which type of conservative intervention is superior, several studies have proved that an ESI is able to increase patients’ quality of life, relieve lumbosacral radicular pain and finally, reduce or delay more invasive interventions, such as spinal surgery. Although ESIs should represent a treatment of choice in the case of acute LBP or leg pain, in our research we focused on the efficacy of ESIs in the treatment of chronic LBP.

The aim of this narrative review is to analyze the mechanism of action of ESIs in chronic lumbar pain patients and to understand their current use, application and success in treating this significant widespread pathology. 

## 2. Materials and Methods

A literature review using online databases was carried out regarding the use of epidural steroid injections for lumbar canal stenosis and disk herniation. Articles were extracted from PubMed, Google Scholar, MEDLINE, UpToDate, Embase and Web of Science, combining the terms “spinal disease,” “radicular pain,” “spinal stenosis,” “canal stenosis,” “disk herniation” and “epidural steroid injection” as keywords for the research. Only papers in the English language and regarding human studies were taken into consideration. Non-English language studies were excluded. Scientific publications up to September 2021were included. Only papers focusing on epidural steroid injections for lumbar canal stenosis or disk herniation were included. All reference lists of the relevant studies were then screened to identify any missing publications. The search and the study selection were performed by two investigators (G.P.; A.S.) working independently. At the first level, the titles and abstracts of identified studies were screened. At the second level, the full texts were retrieved and assessed. Ethical approval and patient informed consent were not required because this was a review of previously published studies and did not involve direct contact with patients or alterations to patient care. Any discrepancies were resolved by a third author (M.C.) through consensus. The following data were extracted from each eligible study: first author’s name; publication year; study design; intervention protocol type (the type and amount of steroid and local anesthetic used for the ESI and therapies or medication used for conservative treatment); outcome parameters including Visual Analogic Scale (VAS), Numeric Rating Scale (NRS), Oswestry Disability Index (ODI) and successful events; and the summary of findings.

## 3. Results

### 3.1. Mechanism of Lumbosacral Radicular Pain

Low back pain and radicular pain are caused by interrelated biomechanical and biochemical factors. With the advancement of age and the presence of chronic diseases such as diabetes, obesity, smoking and overload, a series of degenerative processes occur inside the intervertebral disk [17]. The intervertebral disk is approximately 7 to 10 mm thick and 4 cm in diameter and is formed of two different components: the nucleus pulposus, rich in water and glycopeptides, and the anulus fibrosus, constituted of a series of 15 to 25 rings, or lamellae, with collagen fibers parallel to the lamellae in addition to elastin fibers. A thin hyaline cartilage endplate is the interface between the disk and the superior and inferior vertebrae bodies. When the nucleus becomes less elastic and the anulus less continent due to aging, dehydration, inflammatory conditions and/or prolonged misusage of the back, a part of the nucleus can herniate, usually backward. This causes an inflammatory state in the epidural space and the increase in cytokines and other inflammation mediators. This condition, on the one hand favors the ulterior herniation of the nucleus pulposus and on the other, it compresses and stimulates the spinal nerve roots, resulting in back and radicular pain [18,19].

Generally, sciatica from lumbar disk herniation is a self-limiting condition that improves in weeks or months without medical intervention; in some cases, rest, analgesic drugs and a structured exercise program may be needed. Usually, the inflammatory state is more important than the mechanical compression in the pathogenesis and the chronicity of the disease, unless there are no neurological deficits [20]. However, in patients who are refractory to conservative treatment, surgery is usually recommended.

Lumbar spinal canal stenosis is a process that could be part of the aging process and can be related to herniation of an intervertebral disk. Other common causes of stenosis are: congenital deformities; spondylolisthesis; osteophytes; arthritic degeneration; synovial cysts; hypertrophy of the facet joints; hypertrophy of the ligamentum flavum; epidural lipomatosis; spondylosis of the intervertebral disk margins; previous surgery; and neoplastic diseases. All these factors could cause lumbar nerve root compression with microvascular ischemia, axonal injury, intraneural fibrosis and an inflammatory state, leading to chronic back pain [21]. 

### 3.2. Rationale of Epidural Steroid Injections

Epidural injections are performed using a Tuohy needle with the tip placed inside the epidural space, which is located between the ligamentum flavum and the dura mater. Usually, the epidural space is localized thanks to the loss of resistance (LOR) technique, where the needle is advanced between the spinal processes of the vertebras with the help of a syringe full of air or saline solution, which is used to continuously test the pressure on the piston of the syringe. The needle passes through the ligamentum flavum and, when the epidural space is reached, a loss of resistance is felt by the operator on the syringe piston. Moreover, epidural injections can also be performed rapidly under CT and navigation guidance (Figure 1). These techniques can be used to precisely guide needle placement, allowing for the visualization of the optimal needle path and identification of potential problems, such as narrow intralaminar spaces and spinal stenosis, before needle insertion (Figure 2).

Corticosteroids are widely used in regional anesthesia and chronic pain procedures, such as epidural injections, intraarticular injections and nerve blocks. Corticosteroids have a similar structure and activity to the endogenous produced hormone cortisol, which has an anti-inflammatory, immunosuppressive, vasoconstrictive and antiproliferative effect. They work by preventing the enzyme PLA2 from liberating arachidonic acid from the cells. This inhibits the cyclo-oxygenase and lipoxygenase production, which is responsible for the level of prostaglandins, thromboxanes and leukotrienes, before finally decreasing the inflammatory state [22]. They also inhibit the nerve transmission in nociceptive C fibers and reduce vasal permeability, which decreases intraneural and perineural oedema. Local anesthetics have been administered in the epidural space since 1901; however, the epidural use of corticosteroids has only been documented since 1952 [23]. Their efficacy when administrated via epidural injections has been demonstrated in various studies and a stronger effect has been proven in patients with a higher protein count in the cerebrospinal fluid, which is usually associated with an inflammatory state [24]. The prolonged use of corticosteroids at high doses has many systemic side effects and can also result in iatrogenic adrenal gland suppression; however, the epidural administration limits the systemic side effects because a smaller dose is necessary to reach the pharmacological target and its diffusion into systemic circulation is more difficult than in other types of administration [25]. In the Yang et al. [26] meta-analysis regarding lumbosacral radicular pain due to any cause, the use of ESIs resulted in the more effective in control of lumbosacral pain compared to pure conservative treatment, both in short and intermediate terms. However, two other recent meta-analyses have shown a similarity in efficacy and duration, in terms of pain reduction and functional gain, between local anesthetic alone or local anesthetic with a corticosteroid epidural injection [27,28].

### 3.3. Epidural Steroid Injections for Disk Herniation Lumbar Pain

The epidural administration of corticosteroids is one of the most common mini-invasive medical treatments for chronic spinal pain caused by disk herniation [29,30]. In fact, in the absence of chronic severe pain or neurological deficit, epidural steroid injections may be the treatment of choice for disk herniation. As mentioned before, it reduces the concentration of inflammatory mediators in the epidural space and vascular permeability [31]; it also reduces the damage of C fibers, which diminishes the pain [32,33]. In particular, the anesthetic effect of methylprednisolone over other steroids and non-steroid anti-inflammatory drugs has been proven when injected into the epidural space [34]. 

In clinical practice, both corticosteroids and local anesthetics are used [35,36]; the former are used to reduce the inflammation for a prolonged time, while the latter are used to mitigate the discomfort of the procedure and immediately decrease pain.

The difference in the use of a local anesthetic alone or local anesthetic with a corticosteroid in the epidural administration to treat disk herniation pain has been indagated in numerous papers without a clear result [37,38], although a meta-analysis written by Lee at al. evidenced a small difference between the epidural injection of lidocaine and lidocaine plus corticosteroids [39]. 

However, a good number of studies have described significant pain relief and improvement of functional status after an ESI, especially in short–medium terms [40,41,42,43,44,45] (Table 1). In fact, Kennedy and colleagues have found a high rate of success of ESIs at 6 months in their study, but there was also a recurrence of the symptoms during the 5 years follow-up after the injection [46]. In a similar way, Buchner et al. found a significant improvement in patients treated with epidural steroid injections for a very short period after the treatment but no improvement was seen after 6 weeks and 6 months, compared to the control group who did not receive the injection [47]. A response to the treatment after 1 h of having the procedure has been suggested as predictive for favorable medium-term success [48]. Interestingly, Buttermann et al. suggested that ESIs could be more effective in patients who presented magnetic resonance imaging of inflammatory endplate changes [49]. 

On the other hand, the Spine Patient Outcomes Research Trial (SPORT) [50], a prospective multi-center study of the operative versus nonoperative treatment of lumbar intervertebral disk herniation, found no improvement in short- or long-term outcomes in patients who received ESIs compared to patients who did not. However, it is important to say that an increased rate of surgical avoidance was observed in the group treated with ESIs; this could underline the role of conservative treatments, also considering the high incidence of the spontaneous reabsorption of lumbar disk herniation (66.66% according to Zhong et al.) [51].

Finally, Kreiner et al. [52], in their guideline for the diagnosis and treatment of disk herniation with radicular pain, stated that an ESI is indicated for a proportion of patients with lumbar disk herniation to provide symptom relief in the short term (2–4 weeks) with a grade A recommendation. Additionally, at the moment, no sufficient evidence exists to make a recommendation regarding the 12-month, or more, efficacy of ESIs.

**Table 1 ijerph-19-00231-t001:** Epidural steroid injections for disk herniation studies.

Author, Year	Study Design	Study Protocol		Outcome Measures	Summary of Findings
		Steroid Injection	Control		
Sariyildiz MA, 2017 [40]	Prospective (repeated measures)	Transforaminal, 40 mg betamethasone + lidocaine 2%	Baseline	VAS, Oswestry Disability Index (ODI), hospital anxiety and depression scale, and Pittsburgh Sleep Quality Index (PSQI)	Compared to baseline measurements, there were significant improvements (> 50%) in radicular pain, ODI, depressive symptoms and PSQI scores at two weeks and 12 months after injection
Guclu B, 2020 [41]	Prospective (repeated measures)	Transforaminal 3 mL 0.33% lidocaine + 4 mg dexamethasone	Baseline	VAS scores at 12 weeks	Transforaminal epidural steroid injection is effective in relieving radicular pain, especially in paramedian lumbar disk herniation
Kennedy DJ, 2017 [46]	Prospective	Transforaminal epidural steroid injection	Baseline	Presence of recurrent or persistent pain, pain within the previous week, current opioid use for radicular symptoms, need for additional spinal injections, progression to surgery and unemployment due to pain	Despite a high success rate at 6 months, the majority of subjects experienced a recurrence of symptoms at some time during the subsequent 5 years. Few reported current symptoms and a small minority required additional injections, surgery or opioid pain medications
Manchikanti L, 2014 [29]	RCT, double-blind	Transforaminal 1% lidocaine, followed by 3 mg or 0.5 mL betamethasone	1.5 mL 1% lidocaine + 0.5 mL sodium chloride	Numeric Rating Scale (NRS), Oswestry Disability Index 2.0 (ODI), opioid intake	At 2 years, there was significant improvement in all participants, although there was a lack of evidence of the superiority of steroids compared to local anesthetic
Manchikanti L, 2014 [30]	RCT, double-blind	Interlaminar 0.5% lidocaine (6 mL) + 1 mL betamethasone	0.5% lidocaine (6 mL)	Numeric Rating Scale (NRS), Oswestry Disability Index 2.0 (ODI), opioid intake	Improvement in 70% of the steroid group and 60% of the control group at the end of 2 years.
Manchikanti L, 2013 [31]	RCT, double-blind	Interlaminar 0.5% lidocaine (5 mL) + 1 mL betamethasone	0.5% lidocaine (6 mL)	Pain relief and functional status improvement of ≥ 50%	Average relief of 33.7 ± 18.1 weeks in the local anesthetic group and 39.1 ± 12.2 weeks in the local anesthetic and steroid group
Buchner M, 2000 [47]	RCT	Interforaminal 100 mg methylprednisolone in 10 mL bupivacaine 0.25%	10 mL bupivacaine 0.25%	VAS, straight leg raising test and functional status	No significance on pain relief, improvement of straight leg raising and improvement of functional status at 6 weeks and 6 months
Vad VB, 2002 [44]	RCT	Transforaminal epidural steroid injection	Saline trigger-point injection	VAS, patient satisfaction scale, Roland–Morris low back pain questionnaire	At 1.4 years, the group receiving transforaminal epidural steroid injections had a success rate of 84%, vs. 48% for the control group
Butterman GR, 2004 [49]	Prospective	Epidural steroid injection	Baseline	VAS, Oswestry Disability Index [ODI], pain diagram	At 2 years, it was beneficial for a small number of patients with advanced disk degeneration and chronic low back pain. It was more effective in discogenic inflammation
Manchikanti L, 2008 [45]	RCT	Caudal epidural injections with 9 mL 0.5% lidocaine mixed with 1 mL steroid (6 mg betamethasone or 40 mg methylprednisolone)	Caudal epidural injections with 0.5% lidocaine 9 mL	NRS, ODI, opiod intake	Comparable efficacy in both groups at 12 months
Radcliff K, 2012 [50]	Prospective	Epidural steroid injection	No epidural steroid injection	VAS, ODI, patient satisfaction	No improvement in short- or long-term outcomes (4 years) compared to patients who were not treated with ESIs
Şencan S, 2020 [48]	Retrospective	Transforaminal epidural steroid injection	Baseline	NRS	A decreased pain scores at 1 h is a predictor for a favorable 3-month response to an ESI

LDH = lumbar disk herniation; VAS = visual analogue scale; FU = follow-up; ESI = epidural steroid injections.

### 3.4. Epidural Steroid Injections for Canal Stenosis Lumbar Pain

The administration of steroids via epidural injection as a nonsurgical treatment for lumbar spinal stenosis (LSS) has been analyzed in various studies but, again, there is not a clear consensus regarding their efficacy in relieving the symptoms, especially in the long-term follow-ups. It is important to notice that a consistent number of studies have reported some degree of benefit, especially regarding short-term improvements [53,54,55,56,57,58,59] (Table 2). A more favorable response seems to be associated with relative youth, female sex and patients with single level stenosis, while BMI, MRI severity and the dimension of the spinal canal are probably not predictive [55,60,61]. Additionally, individual pain sensitivity does not seem to influence the outcome of an ESI in the patients affected by LSS [62]. Interestingly, Milburn and colleagues, in a randomized study, suggested that the response to the treatment is maximized when the ESI is performed at the intervertebral level of maximal stenosis [63], and their result was confirmed by the trial conducted a few years later by Bajpai et al. [64].

A randomized, double-blind controlled trial with a 2-year follow-up was conducted by Manchikanti et al., which compared the epidural injection of local anesthetic alone to local anesthetic plus steroids, and the authors found a significant relief of the symptoms in a convincing percentage of the patients treated, but without significant difference between the two groups [65]. Accordingly, another large randomized trial on 400 patients, conducted by Friedly and colleagues, found minimal or no short-term benefits in adding steroids to a local anesthetic epidural injection for the treatment of LSS [66].

Moreover, some other studies did not find any significant improvement in symptoms or quality of life after an ESI for the treatment of LSS [67,68,69,70]. Tran et al. wrote a review regarding the nonsurgical treatment of LSS and concluded that the literature could provide only limited evidence to formulate recommendations pertaining to the nonsurgical treatment of LSS [71]. 

Finally, Liu et al., in their systematic review and metanalysis, also concluded that there is minimal evidence to show that epidural steroids are better than local anesthetic alone in the treatment of LSS patients [72].

**Table 2 ijerph-19-00231-t002:** Epidural steroid injections for canal stenosis studies.

Authors, Year	Study Design	Study Protocol		Outcome Measures	Summary of Findings
		Intervention	Control		
Sabbaghan S et al., 2020 [53]	Retrospective, single arm	Bupivacaine hydrochloride 0.5% (3 mL) + triamcinolone acetonide 80 mg (2 mL)	Baseline	VAS for lumbar pain, VAS for lower limb pain and ODI	Improvement in pain (both lumbar than lower limb) and ability
Park CH et al., 2014 [54]	Prospective, single arm	2 mg preservative-free ropivacaine + 1500 units hyaluronidase + 40 mg triamcinolone acetonide	Baseline	5-point patient satisfaction scale at 2 and 8 weeks post-treatment	The ESI seems to provide effective short-term pain relief from LSS. The grade of LSS appears to have no effect on the degree of pain relief
Cosgrove JL et al., 2011 [55]	Prospective, single arm	Steroids, not specified	Baseline	Self-reported activity level and measured walking ability using the SSSQ and SMWT. The results were correlated through demographic data, magnetic resonance imaging (MRI) characteristics and electrodiagnostic results	Significant improvement as measured by changes in SMWT and SSSQ. Relative youth and female sex are associated with a more favorable response
Farooque M et al., 2017 [56]	Case series	10 mg dexamethasone (1 mL) + an equal volume of 2% preservative-free lidocaine on each side (transforaminal)	Baseline	Pain score and Swiss Spinal Stenosis score at baseline, 1, 3 and 6 months	Reduction of ≥ 50% in numeric pain scale score in 30% of participants at 1 month, 53% at 3 months and 44% at 6 months. Swiss Spinal Stenosis subscale scores indicated a significant reduction in the proportion of participants reporting the presence of severe pain in the back, buttocks and legs during FU compared to baseline (*p* < 0.05)
Hammerich A et al., 2019 [57]	Randomized parallel-group trial	1.5 mL steroid (not specified) at each site injected. Reassess at 3–4 and 6–8 weeks for potential second and third injections	1.5 mL of steroid (not specified) at each site injected. Reassess at 3–4 and 6–8 weeks for potential second and third injections + physical therapy (PT)	Disability, pain, quality of life and global rating of change were collected at 10 weeks, 6 months and 1 year, and then analyzed using linear mixed model analysis	The ESI plus PT was not superior to ESI alone for reducing disability in individuals with LSS.
Brown LL et al., 2012 [58]	Randomized controlled trial	80 mg triamcinolone acetate (40 mg for diabetic patients) mixed with 6 mL preservative-free saline injected in divided doses at the treated levels	MILD procedure: a minimally invasive posterior lumbar decompression performed fluoroscopically through a small 6-gauge port	Visual Analog Scale, Oswestry Disability Index and Zurich Claudication Questionnaire (ZCQ) for patient satisfaction	MILD procedure was superior compared to ESI in pain reduction and the improvement of functional mobility
Kim HJ et al., 2013 [62]	Prospective, single arm	40 mg (1 mL) triamcinolone acetonide suspension + 1 mL bupivacaine hydrochloride 0.5% + 1 mL of saline	Baseline	Pain sensitivity questionnaire (PSQ), Oswestry Disability Index (ODI), and Visual Analog Scale (VAS) for back and leg pain	Significant decreases in VAS for back/leg pain and ODI 2 months after ESI. Individual pain sensitivity does not influence the outcomes of ESI treatment in patients with LSS
Campbell MJ et al., 2007 [60]	Controlled clinical trial	Steroids (not specified) once a week for 3 weeks	Baseline	Spinal canal dimension, resolution of symptoms after ESI, necessity of surgery after ESI	Spinal canal dimension is not predictive of the success or failure of ESIs in patients with LSS
Milburn J et al., 2014 [63]	Randomized controlled trial	2 mL 40 mg/mL methylprednisolone + 2 mL bupivacaine 0.25% + 2 mL normal saline at the most stenotic level	2 mL 40 mg/mL methylprednisolone + 2 mL bupivacaine 0.25% + 2 mL normal saline at 2 intervertebral levels cephalad of the level of maximal stenosis	Analog pain scale from 0 to 10 during ambulation and at rest, Roland–Morris Disability Questionnaire (RDQ) at baseline, immediately after ESI and at 1, 4, and 12 weeks post-injection	Symptom improvement is optimized when the ESI is performed at the intervertebral level of maximal stenosis
Bajpai S et al., 2020 [64]	Randomized controlled trial	5 mL bupivacaine (0.25%) + 2 mL methylprednisolone acetate (40 mg/mL) + 1 mL normal saline at maximal stenotic intervertebral level	5 mL bupivacaine (0.25%) + 2 mL methylprednisolone acetate (40 mg/mL) + 1 mL normal saline 2 intervertebral levels cephalad to the level of maximal stenosis	Numeric Pain Rating Scale (NPRS) and Oswestry Disability Index (ODI) at 2, 6 and 12 weeks after the intervention	Pain relief improvement is optimized when the ESI is performed at the maximum stenotic intervertebral level
Koc Z et al., 2009 [59]	Randomized controlled trial	Group 1: inpatient physical therapy program for 2 weeks Group 2: 60 mg triamcinolone acetonide + 15 mg 0.5% bupivacaine hydrochloride + 5.5 mL saline	Group 3: no intervention	Pain severity by Visual Analog Scale (VAS), Finger Floor Distance (FFD) (cm), Treadmill Walk Test, Sit-to-Stand Test (Seconds), Weight-Carrying (WC) Test (Seconds)	Both ESI and physical therapy groups demonstrated an improvement in symptoms and in outcomes measured without any significant differences
Manchikanti L et al., 2015 [65]	Randomized controlled trial	Local anesthetic (lidocaine 0.5%) 5 mL mixed + 6 mg betamethasone (1 mL)	6 mL local anesthetic (lidocaine 0.5%)	Numeric Pain Rating Rcale (NPRS) and Oswestry Disability Index (ODI) at 3, 6, 12, 18 and 24 months post-treatment	Epidural injections of local anesthetic with or without steroids provide relief in a significant proportion of patients with LSS
Shamov T et al., 2020 [61]	Prospective, two arms	Group 1: 10 mg dexamethasone in 3 cc 0.25% bupivacaine for patients with discogenic sciatica Group2: 10 mg dexamethasone in 3 cc 0.25% bupivacaine for patients with LSS		Pain intensity was assessed by VAS at baseline and on days 1, 15 and 30 after intervention	ESIs are more effective in patients with discogenic sciatica than in single level LSS. In multiple level LSS, results are disappointing
Friedly JL et al., 2014 [66]	Randomized controlled trial	1 to 3 mL 0.25% to 1% lidocaine followed by 1 to 3 mL triamcinolone (60 to 120 mg), betamethasone (6 to 12 mg), dexamethasone (8 to 10 mg) or methylprednisolone (60 to 120 mg)	1 to 3 mL 0.25% to 1% lidocaine alone	Roland–Morris Disability Questionnaire (RDQ) and the rating of leg pain intensity (on a scale from 0 to 10) at 6 weeks after injection	Epidural injections of glucocorticoids plus lidocaine offered minimal or no short-term benefits compared to epidural injections of lidocaine alone
Makris UE et al., 2017 [66]	Subsequent analysis of a randomized controlled trial (Friedly JL et al., 2014 [58])	1 to 3 mL 0.25% to 1% lidocaine followed by 1 to 3 mL triamcinolone (60 to 120 mg), betamethasone (6 to 12 mg), dexamethasone (8 to 10 mg) or methylprednisolone (60 to 120 mg)	1 to 3 mL 0.25% to 1% lidocaine alone	RDQ (Roland–Morris Disability Questionnaire), Sickness Impact Profile (SIP) weights assigned to the RDQ items and patient-prioritized RDQ items at 6 weeks after injection	No significant difference between groups for the RDQ or patient-prioritized RDQ, and while the difference between groups for RDQ using SIP weights was statistically significant, this was not clinically important
Tomkins-Lane CC et al., 2012 [68]	Prospective, single arm	Steroids, not specified	Baseline	Total activity (performance) measured over 7 days and maximum continuous activity (capacity), walking capacity was also assessed with the Self-Paced Walking Test and subjects completed the ODI, SSSQ, Medical Outcomes Study 36-Item Short-Form Health Survey, Visual Analog Pain scales and body diagrams	At 1 week postinjection, 58.8% of the subjects demonstrated increased total activity and 53% had increased maximum continuous activity, although neither change was statistically significant. Patients perceived improvements in symptoms, but these were not reflected in significant changes in performance or capacity
Rivest C et al., 1998 [69]	Prospective, two arms	Group 1: 3 mL 0.5% lidocaine and 3 mL methylprednisolone acetate, followed by 3 mL saline for patients with LSS Group 2: 3 mL 0.5% lidocaine and 3 mL methylprednisolone acetate, followed by 3 mL saline for patients with discogenic sciatica		Pain was assessed at baseline and 2 weeks following a single ESI using a Visual Analog Scale	LSS patients have worse responses to ESIs than herniated disk patients
Fukusaki M et al., 1998 [70]	Randomized controlled trial	Group 3: 8 mL 1% mepivacaine and 40 mg methylprednisolone	Group 1: 8 mL saline Group 2: 8 mL 1% mepivacaine	Evaluation of improvement on pseudo-claudication associated with LSS as follows: excellent effect, > 100 m in walking distance; good effect, 20–100 m in walking distance; poor effect, < 20 m in walking distance	ESIs have no beneficial effects on walking ability associated with LSS compared to epidural injections with a LA alone

ESI = epidural steroid injection; LSS = lumbar spinal stenosis; LA = local anesthetic; FU = follow-up; VAS = Visual Analog Scale; ODI = Oswestry Disability Index; SSSQ = Swiss Spinal Stenosis Questionnaire; SMWT = 6-Minute Walk Test; MRI = magnetic resonance imaging; PT = physical therapy; MILD = minimally invasive lumbar decompression; ZCQ = Zurich Claudication Questionnaire; PSQ = Pain Sensitivity Questionnaire; FFD = Finger Floor Distance; SIP = Sickness Impact Profile.

### 3.5. Image-Guided Epidural Injections

Special consideration should be given to image-guided techniques that may help the clinician in performing epidural injections, although no significant difference has been shown regarding outcomes. The use of ultrasound in performing an interlaminar approach could help to estimate the distance from the skin to the epidural space and the optimal needle direction [73,74]. However, although it is a radiation-free technique, ultrasound is limited by the operator’s experience and the real-time visualization of needle tip advancement could be challenging, especially in obese patients [75]. Fluoroscopy (x-rays) and computerized tomography (CT) have both proved to be effective and safe techniques for guiding transforaminal epidural injections, although the former provides less radiation exposure for patients [76,77]. Moreover, new generation CT devices may integrate neuronavigation systems that are able to perform a computerized analysis in order to best define the needle’s path towards the epidural space. Its use has been described for spinal surgery but it may be expanded to transforaminal epidural injections as well, although the high cost of these devices should be considered [78]. Future studies are expected to determine the best technique in terms of efficacy and safety for both patients and clinicians.

## 4. Discussion

Chronic lumbar pain is a widespread problem, which affects a large part of population at some point of their life. Disk herniation and canal stenosis are the most common causes and they need to be treated due to the high impact of the symptoms on patients’ quality of life, especially because they could affect walking and ability to work [18]. 

Surgical intervention has been proven to be effective but is not usually considered as the first option [7]. On the other hand, nonsurgical treatments, such as epidural steroid injections, do not have clear literature consensus. Fully understanding whether ESIs would be able to relieve symptoms and delay or prevent surgery could be a crucial step, especially because chronic back pain patients are typically elderly and multimorbid who could be more affected by the impact of surgery. 

In this narrative review, we tried to analyze the existing literature regarding the use ESIs for these kinds of patients, considering randomized controlled studies as well as reviews, meta-analyses and guidelines. 

Overall, ESIs seem to be safe and quite effective in relieving the main symptoms, especially in short-term follow-ups, and in delaying surgery, according to a consistent number of studies and to the guideline written by Kreiner and colleagues [52]. Moreover, ESIs could be more powerful in the case of patients with disk herniation than patients with canal stenosis [61]. Attention should also be paid to the technique and to the vertebral level of the injection, at least for spinal stenosis [63,64].

As mentioned before, literature consensus is still missing and numerous studies did not find significative improvements, especially in long-term follow-ups. In addition, it seems to be difficult to find significative differences between using local anesthetics alone or local anesthetics plus steroids in the injection. 

Due to the anti-inflammatory action of steroids, patients with high local inflammatory status could probably benefit more from using steroids [49]. However, epidural steroid injections have also been associated with potential adverse effects, including acute neurological symptoms [79,80], in addition to other possible complications related to the epidural technique, i.e., inadvertent dural puncture, hematomas and infection. [81]

In this context, the identity of the patients who could benefit the most from this procedure has not been completely established yet and could be a crucial future goal. 

This narrative review has some limitations. Firstly, the studies taken into consideration did have different epidural injection approaches. Secondly, the heterogeneity of the enrolled patients, analyzed parameters and data collection in the studies taken into consideration could be an important bias. Lastly, the heterogeneity of the purposes of the studies, such as the comparison of steroid injections versus nothing or local anesthetic injections versus local anesthetic plus steroids, could make global analysis difficult.

Surely, more randomized studies with larger numbers of patients are needed to fully understand the efficacy of ESIs and define which patients could benefit more from the procedure, especially in order to delay or prevent surgery.

## 5. Conclusions

According to the literature analyzed in this narrative review, there is no consensus on the use of ESIs for patients with chronic lumbar pain. ESIs seem to be effective in relieving symptoms in the short term and delaying surgery, while evidence of any long-terms benefits is still lacking. More studies are needed to better understand which patients could benefit more from epidural steroid injections.

## Figures and Tables

**Figure 1 ijerph-19-00231-f001:**
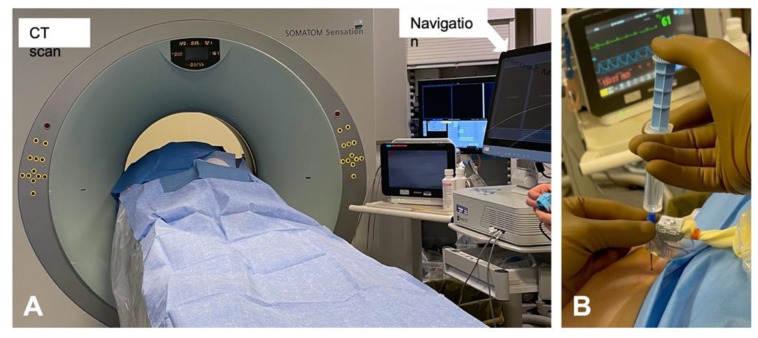
Representative images of (**A**) intraoperative setting for CT and navigation guided epidural injection and (**B**) navigated needle insertion.

**Figure 2 ijerph-19-00231-f002:**
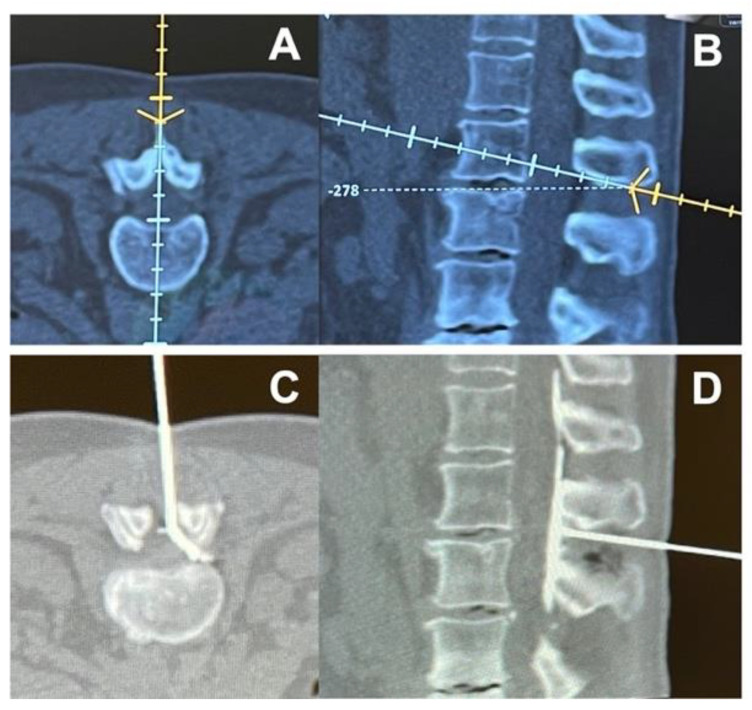
Representative images of (**A**,**B**) navigated planning of needle path to the epidural space and (**C**,**D**) evidence of injection in the epidural space by the use of a contrast agent.

## Data Availability

Not applicable.

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
