# Peer review of "Epidural Steroid Injections for Low Back Pain: A Narrative Review"

_ijerph, 2021, doi:10.3390/ijerph19010231_

Round 1

Reviewer 1 Report

First of all, well structured and great review. 

Author should clearly discriminate between chronic back pain and acute back pain, back pain and leg pain. I understand this study is focused on chronic lumbar pain rather than acute pain and leg pain, I would request authors to clearly address that they are focused on back pain and not mentioning about leg symptom. In my opinion, lumbar disc herniation is usually acute condition, and degeneration process after disc herniation promotes degenerative lumbar spondylosis. Additionally, ESI could potentially used for acute conditions and leg symptom; author should also mention about that. 

L105  more important than the mechanical compression in the pathogenesis,,” unless there are no neurological deficits”. 

L111  additional causes of lumbar spinal canal stenosis: spondylolisthesis, epidural lipomatosis, 

L114 All these factors do potentially cause chronic back pain finally, however, again, author should clearly discriminate between chronic back pain and acute back pain, back pain and leg pain. 

Author Response

Thank you very much for your appreciation.

Best regards.

Reviewer 2 Report

In this narrative review, the authors summarized the application of epidural steroid injection for low back pain management; outlined the mechanism of lumbosacral radicular pain symptom, rationale of epidural steroid injection and the indications; discussed the safety and efficacy of epidural steroid injection. This study is well designed, and the data from literatures is well interpreted. While some new findings can be recognized, some minor issues are raised as well.

  • Regarding the procedure is performed in patients with low back pain caused by lumbar disk herniation and/or lumbar canal stenosis with anatomical abnormity, image guided operation becomes essential for the outcome. The authors can take this into account and analyze the results of the procedure guided by X-ray, computerized tomography, neuronavigation and ultrasound respectively.
  • The safety and efficacy of epidural steroid injection is well discussed in the manuscript; however, the complication of the procedure was missed. Recent reports indicate that local epidural injection can led to severe consequences (PMID: 28824864, PMID: 10969325). The authors can add some inputs about the potential complications for the epidural steroid injection for low back pain in the discussion section

Author Response

(The authors gave the same response as above.)

Round 2

Reviewer 1 Report

Acceptable as is.